# Arsenic in Drinking Water and Diabetes

Aryatara Shakya [1], Matthew Dodson [1], Janick F. Artiola [2], Monica Ramirez-Andreotta [2], Robert A. Root [2], Xinxin Ding [1], Jon Chorover [2] and Raina M. Maier [2,*]

1   Department Pharmacology & Toxicology, University of Arizona, Tucson, AZ 85721, USA;
    aryatarashakya@arizona.edu (A.S.); dodson@pharmacy.arizona.edu (M.D.);
    xding@pharmacy.arizona.edu (X.D.)
2   Department Environmental Science, University of Arizona, Tucson, AZ 85721, USA;
    artiola@gmail.com (J.F.A.); mdramire@arizona.edu (M.R.-A.); robroot.az@gmail.com (R.A.R.);
    chorover@arizona.edu (J.C.)
*   Correspondence: rmaier@ag.arizona.edu

**Abstract:** Arsenic is ubiquitous in soil and water environments and is consistently at the top of the Agency for Toxic Substances Disease Registry (ATSDR) substance priority list. It has been shown to induce toxicity even at low levels of exposure. One of the major routes of exposure to arsenic is through drinking water. This review presents current information related to the distribution of arsenic in the environment, the resultant impacts on human health, especially related to diabetes, which is one of the most prevalent chronic diseases, regulation of arsenic in drinking water, and approaches for treatment of arsenic in drinking water for both public utilities and private wells. Taken together, this information points out the existing challenges to understanding both the complex health impacts of arsenic and to implementing the treatment strategies needed to effectively reduce arsenic exposure at different scales.

**Keywords:** arsenic; diabetes; drinking water treatment; arsenic exposure





## 1. Introduction—Water Quality and Importance

Safe and affordable drinking water is a prerequisite for prosperity and sustainable development. The 2021 World Economic Forum report [1] lists natural resources crises, which includes water, as the fifth-highest existential threat globally. According to the 2017 WHO and UNICEF reports, more than 785 million people that year did not have access to basic water services [2]. While substantial progress has been made worldwide to provide access to clean drinking water, many regions have limited surface-water supplies and rely on groundwater resources. This has led to an increased risk of developing health issues in many parts of the world [3]. Metal(loid)s such as zinc (Zn), selenium (Se), copper (Cu), molybdenum (Mo), chromium (Cr), manganese (Mn), nickel (Ni), cobalt (Co), iron (Fe), magnesium (Mg), and arsenic (As) rank among the top priority metals that act as environmental toxicants in drinking water worldwide [4–10]. This review focuses in particular on As, a ubiquitous element that has been at the top of the Agency for Toxic Substances Disease Registry (ATSDR) substance priority list since 1997 [11], as it has been shown to induce toxicity even at low levels of exposure, thus representing a continuously growing public health concern.

Exacerbating the issue is the fact that exposure are not equal. Environmental racism and injustices result in people of color and low-income community members living in closer proximity to sources of environmental pollution (e.g., [12,13]). As a case in point, on 31 May 2022, the Biden–Harris Administration established a Department of Health and Human Services Office of Environmental Justice. As stated in the press release by HHS Secretary Xavier Becerra: "The blunt truth is that many communities across our nation—particularly low-income communities and communities of color—continue to bear

the brunt of pollution from industrial development, poor land use decisions, transportation, and trade corridors" [14]. Social determinants of health (SDH), the nonmedical factors that influence health, are a primary indicator of one's health and can account for 30–55% of health outcomes [15]. SDH factors can impact arsenic exposure, and in turn, influence the incidence of disease and morbidity. These factors are related to economic stability, education, health and health care, neighborhood and build environment, and social and community contexts and include, for example, access to healthy foods, quality of housing and infrastructure, environmental conditions, civic participation, and early childhood development [15,16]. These aforementioned SDH and others are influencing health disparities and inequities, thus creating vulnerabilities—the degree to which people and places can be harmed due to external stresses on human health (e.g., [17–20].

This review presents current information related to the distribution of arsenic in the environment, the resultant impacts on human health, especially related to diabetes, regulation of arsenic in drinking water, and approaches for treatment of arsenic in drinking water for both public utilities and private wells. Understanding these different perspectives is important for the prevention and mitigation of arsenic exposure. This review is presented in the context of diabetes—almost half a billion people worldwide live with this disease and the prevalence is projected to continue increasing [21]. The objective of this review is to delineate the existing challenges to understanding both the complex health impacts of arsenic and to implementing the treatment strategies needed to effectively reduce arsenic exposure at different scales.

## 2. Health Impacts of Arsenic in Drinking Water

Arsenic is a naturally occurring ubiquitous metalloid, the inorganic forms of which (iAs) are predominantly found in soil, sediment, and surface and groundwater reservoirs [22]. Depending on the pH, redox state, temperature, and solution composition, arsenic is generally soluble in groundwater [22,23]. Major sources of As contamination in drinking water include waste products from gold mining and mineral extraction, agricultural pesticides, and thermal springs, all of which contribute to As accumulation in groundwater [24]. While the gastrointestinal tract readily absorbs the inorganic forms of arsenic, resulting in their distribution throughout the body, they are mainly metabolized via methylation in the liver by arsenic methyltransferase (AS3MT) to their organic counterparts, namely, monomethylarsonic acid (MMA) and dimethylarsinic acid (DMA), then excreted primarily in urine. More than 200 million people worldwide are exposed to iAs at concentrations above the EPA- and WHO-designated safe limit of 10 μg/L [25]. Based on data compiled from the mid- to late 1990s by the USGS from wells used throughout the US as public drinking water sources, it is estimated that 8% of the public drinking water supply may exceed 10 μg/L [26]. Importantly, consumption of arsenic-contaminated drinking water is associated with numerous disease states, including cancer, cardiovascular disease, skin lesions, nephrotoxicity, neurological disorders, and diabetes [8,27]. As such, investigations on how arsenic promotes disease progression, including diabetes, have garnered much attention over the past few decades, particularly because chronic exposure to arsenic in drinking water has been associated with an increased risk of type 2 diabetes in arsenic-rich areas worldwide [28].

Within the context of diabetes, understanding and mitigating the impact of SDH are priorities. For example, those of lower socioeconomic status are more likely to develop type 2 diabetes mellitus, experience more complications, and die sooner than those of higher socioeconomic status [29]. Furthermore, disadvantaged communities can experience several routes of arsenic exposure that are compounded by SDH factors [30–33]. For example, American Indians/Alaskan Natives (15.9%) and Hispanics (12.8%) have a greater prevalence of diabetes when compared to non-Hispanic whites (7.6%) across the US [34]. In addition to the years of life lost, $237 billion is spent in direct medical costs and $90 billion is lost in reduced productivity due to diabetes [35].

Finally, it is important to note that arsenic exposure can occur from food consumption as well as from drinking water. In general, food exposure is primarily from purchased foods, such as store-bought rice, cereals, and fruit juices. Meat, poultry, dairy products, cereals, and vegetables contain higher proportions of inorganic arsenic forms (e.g., [36–39], and food preparation practices can influence the concentration of arsenic in foods. For example, several studies have highlighted how cooking in arsenic-laden water, specifically boiling foods, such as maize grains, cereals (e.g., rice and quinoa), and vegetables that hold a noteworthy amount of water during boiling, can lead to arsenic exposure via the consumption of the cooked foods [40–43]. Since food type and preparation are tied to culture, place, geography, and race/ethnicity, it is critical to acknowledge how culturally relevant foods and cooking practices can influence individual/family/community arsenic exposure.

## 3. Arsenic Distribution in the Environment

Arsenic is naturally occurring and ubiquitous, distributed in the environment by both natural and anthropogenic processes [44]. It is present, at least in trace amounts, in nearly all crustal rocks and sediments. Arsenic is listed in 7133 minerals, inclusive of nonessential stoichiometries, and occurs as a principal structural constituent in 728 validated mineral species, including elemental arsenic ($As^0$), arsenides ($As^{3-}$), sulfides ($As^{2+,3+,5+}$), oxides ($As^{3+,5+}$), arsenites ($As^{3+}$), and arsenates ($As^{5+}$) [45]. While mineral specimens are rare in nature, arsenic occurs with ore minerals or alteration products, the most important being arsenian pyrite ($Fe(S,As)_2$), arsenopyrite (FeAsS), and scorodite ($FeAsO_4 \cdot 2H_2O$) [46,47]. Since the 1983 discovery of elevated dissolved arsenic in the Bangladeshi tube wells installed for pathogen-free drinking water, there has been widespread recognition of the large-scale global health problems resulting from chronic exposure, which has placed high priority on understanding the mobility, bioavailability, and toxicity of arsenic in the aqueous environment [48–51].

Geological processes concentrate arsenic in the Earth's crust through magmatic and hydrothermal processes, becoming enriched most commonly in chalcophile metallic ore deposits [52]. In depositional systems, arsenic accumulates in aquifer sediments comprising geologically young (Cenozoic) alluvium, commonly hydrologically down-gradient of geothermally and magmatically enriched zones [53]. Leaching of arsenic into drinking water sources results in serious and extensive human and ecosystem health risks. Natural processes that mobilize arsenic to contaminate ground and surface waters from its primary geogenic sources include (i) redox-driven weathering, principally *oxidative* weathering of (arsenian) sulfides and *reductive* dissolution (arsenic sorbed) ferric hydroxides, (ii) volcanism, and (iii) biological activity. Elevated concentrations of arsenic in groundwater aquifers have been observed along the Pacific Ring of Fire [54] and reported in hot spots with arsenic at levels problematic to health in the Bengal delta [55,56], Red River delta (China) [57], Mekong delta (Vietnam) [58], Indus delta (Pakistan) [59], Taiwan [48,60], the western United States [61], Canada [62], and Argentina [63–65]. Anthropogenic activities also mobilize arsenic into the environment from extraction and beneficiation of ore, fossil fuel combustion, and the application of arsenic-containing pesticides, herbicides, and fertilizers [66–68].

Arsenic is unlike many other inorganic contaminants in that processes of environmental biogeochemical cycling in the range of pH and Eh common to the shallow subsurface can alter its speciation, which in turn affect its solid–aqueous phase partitioning [68,69]. That elevated levels of arsenic in groundwater threaten human health in widespread areas is known; however, dissolved arsenic concentrations are commonly spatially unpredictable [56]. The variable character of dissolved arsenic has been attributed to its redoximorphic speciation, electronic structure, and bonding properties, which result in dynamic transformation of its chemical form and phase stability [70]. The processes governing arsenic mobility in aquifers and through sediments are sorption, precipitation, and dissolution. These sequestration and release mechanisms are affected by pH, Eh, and concentrations of competing ions and are generally tied directly to coupled environmental

redox reactions with iron and sulfur [69–72]. Arsenic is removed from the aqueous phase by two primary mechanisms—methylation and subsequent volatilization—and sequestration to the solid phase by (i) sorption at mineral surface sites [73–76], (ii) (co)precipitation with metal (hydr)oxides [69,77,78], or (iii) precipitation as arsenic sulfide under sulfur-reducing conditions [69,79,80]. The reverse reaction of arsenic mobilization is controlled by dissolution of host sulfides or metal hydroxide sorption sites, driven by geochemical redox [81,82]. Recently, nearly 80 studies of arsenic in groundwater around the world, aggregating over 200,000 measurements, were evaluated with machine learning to build a predictive model of arsenic exposure risk [83]. The authors examined 52 environmental variables and found that texture (clay and sand content), pH, and climate showed the greatest statistical importance for predicting elevated dissolved arsenic in aquifers. Model results indicate that 94 to 220 million people are potentially exposed to high levels of arsenic in groundwater, with 85–90% in South Asia.

The solid and aqueous speciation of arsenic directly affects its solubility, mobility, and possibly toxicity [84–86]. In the absence of high sulfide activity, dissolved arsenic in interstitial and surface water is generally present in two oxidation states: arsenite (the trivalent species, $H_xAsO_3^{x-3}$) under suboxic environments, or arsenate (the pentavalent species $H_xAsO_4^{x-2}$) in oxic zones (Figure 1). To a lesser extent, arsenic is found as aqueous organic metabolites [87,88]. Arsenate has dissociation constants of $pKa_1$ of 2.2, $pKa_2$ of 7.0, and a $pKa_3$ of 11.5 [89], and in aerobic waters it is generally found as a combination of the mono- and divalent oxyanions $H_2AsO_4^-$ and $HAsO_4^{-2}$. Under reducing conditions, dissolved arsenic is present as arsenite with $pKa_1 = 9.2$ and $pKa_2 = 13.4$ (Figure 1).

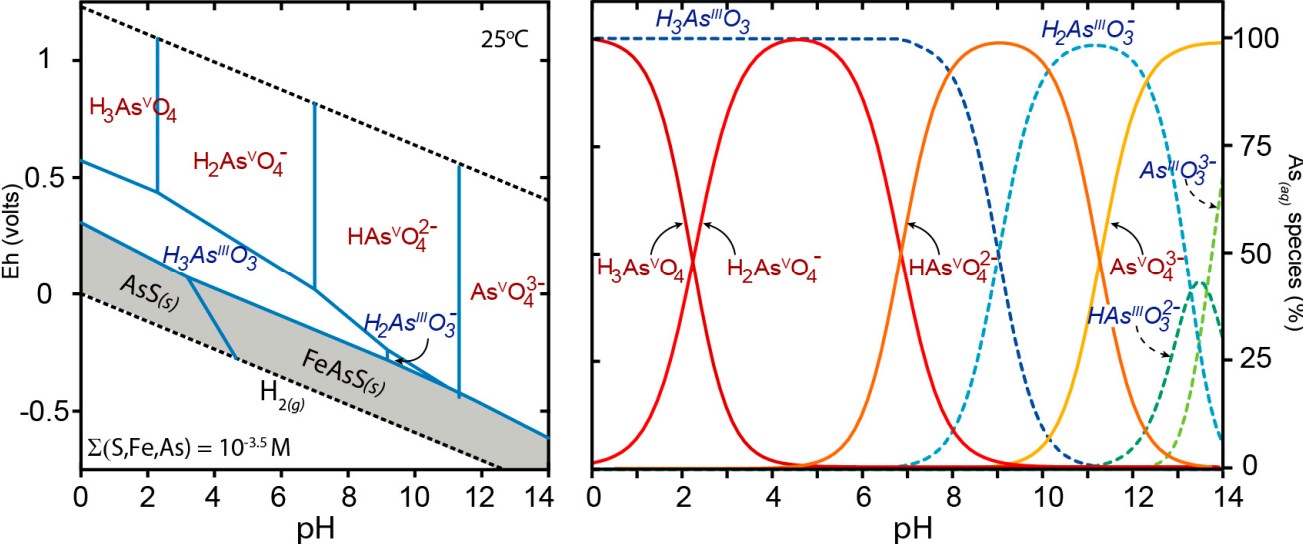

**Figure 1.** Eh-pH activity diagram of arsenic species at 25 °C, 1 bar, (S, Fe, As) = $10^{-3}$ M (**left**). Dashed lines bound the stability field of water, arsenate (As[V]) species are shown in red, arsenite (As[III]) species are shown in blue italics, solid phases are shown with a darkened background. The distribution of pH-dependent dissolved arsenic species are shown (**right**) with arsenate as red solid lines and red text and arsenite in dashed blue lines with blue italic text. Dissolved arsenic species become protonated at low pH and the charge on the oxyanion decreases. Under environmental conditions (pH ≈ 5–9), arsenate generally exists as $H_2AsO_4^-$ and $HAsO_4^{2-}$, while arsenite is the uncharged molecule $H_3AsO_3^0$. Under highly reducing conditions and in the presence of high sulfur activity, solid-phase arsenic sulfides (e.g., AsS) are stable.

Arsenate has been shown to strongly adsorb to positively charged surface sites of metals (oxy)hydroxides and phyllosilicates [73,77,90–92]. Iron, Earth's most abundant redox active element, is commonly found as solid-phase ferric (oxy)hydroxide, which is insoluble under all but very low pH and Eh ranges and exerts strong control over arsenic cycling in the



environment. Positively charged ferric surface coatings in sediments or suspended colloidal particles act as excellent sorbents of oxyanion arsenic (e.g., [75,93]). Therefore, arsenate is significantly immobilized in well-oxygenated sediments rich in iron. When organic matter is broken down through a series of electron transfer reactions in flooded sediments, oxygen is depleted, conditions become suboxic, and redox conditions favor the dissolution of ferric solids [81,94]. This reduction of iron (and arsenic) in suboxic environments is recognized as a primary mechanism of arsenic contamination of groundwater, especially sedimentary aquifers [95].

Groundwater flow, coupled with spatially variant gradients of redox potential and iron and sulfur activities, moves arsenic into and out of solution and thereby through pore spaces in aquifer sediments. Arsenic lability is a function of speciation and the biogeochemical redox characteristics of the subsurface environment controlled by molecular-scale interactions of arsenic at the sediment–water interface. In conditions where microbial activity, including metabolic and detoxification mechanisms, promote a transition from aerobic to anoxic porewaters, arsenate can be reduced to arsenite [88,96]. At the pH of most natural waters, arsenite does not dissociate, is neutral in solution, and the uncharged dissolved species is not as readily adsorbed at metal hydroxide surface sites. Therefore, arsenic phase partitioning in aquifer sediments is generally a function of redox potential and pH [66,71,88,97].

## 4. Diabetes and Arsenic

### 4.1. Diabetes Types and Risk Factors

Not only epidemiological but also a large body of experimental evidence supports the potential role of arsenic in promoting the development of diabetes mellitus (DM). Diabetes mellitus, a metabolic disorder characterized by hyperglycemia and dyslipidemia, is classified into insulin-dependent diabetes mellitus (type 1 diabetes, T1D) and non-insulin-dependent diabetes mellitus (type 2 diabetes, T2D) [98]. T2D, which makes up 90% of all diabetes cases, involves disruptions in whole-body glucose homeostasis due to resistance of peripheral tissue to insulin and decreased insulin production by pancreatic β-cells [99]. In T1D, the immune system destroys the pancreatic β-cells, leading to insulin deficiency [100]. Several toxic metals, such as cadmium, chromium, zinc, mercury, nickel, and arsenic, are known to adversely affect key metabolic pathways, which ultimately plays a role in promoting the development of metabolic disorders, including T1D and T2D [8]. Pathologically, these toxic metals accumulate in the liver, kidney, and pancreas to alter or impair the activity of critical enzymes, organelles, and signaling pathways, leading to adverse effects on metabolism. Critically, these pathological metabolic shifts result in significant increases in blood glucose levels, dyslipidemia, and eventually impaired organ function as a result of constant disruption of physiological homeostasis [101]. While genetics, diet, and lifestyle are established risk factors for developing DM, there is an increased interest in understanding the role of environmental exposure, including arsenic, as a causative factor in driving the diabetes epidemic.

### 4.2. Epidemiological Link between iAs Exposure and Diabetes

The 2011 National Toxicology Program workshop to assess the link between diabetes and the environment found an association between iAs exposure in drinking water and enhanced risk of developing DM, at least at concentrations $\geq$150 μg/L [102]. Epidemiologically, there are several indicators that exposure to iAs in drinking water causes diabetogenic effects. For example, a positive correlation between urinary iAs and its methylated metabolite DMA and increased fasting blood glucose, glycated hemoglobin, and fasting plasma insulin levels, was identified in a patient cohort from northern Mexico. Interestingly, insulin resistance was negatively correlated with iAs exposure in this same cohort, which may shed light on the differential regulation of T2D depending on other confounding variables (i.e., climate, diet, genetic predispositions) [103]. Assessment of the relationship between ingestion of iAs and prevalence of DM in 891 adults in southern Taiwan also showed a pos-

itive correlation between iAs exposure and increased blood glucose levels [104]. Another study conducted in four townships in Taiwan where people consumed iAs-containing well water between the 1900s and 1970s indicated an increase in mortality as a result of diabetes [105]. Reports have also indicated a significant increase in the number of individuals with elevated cholesterol and triglyceride levels in areas with higher iAs concentrations in the drinking water (56 µg/L) compared to an unexposed population (2 µg/L) in Serbia [106]. Furthermore, there was a 9% increase in blood glucose levels (>130 mg/dL) in individuals who consumed iAs-contaminated drinking water for a period of 6 months in Bangladesh [107]. Reports also indicated that a mean iAs concentration of 11 µg/L caused an elevated standardized mortality rate due to diabetic kidney disease and cerebrovascular disease in southeastern Michigan [108].

The number of epidemiological studies examining the relationship between diabetes and arsenic in drinking water has risen in recent years. These studies, which also include follow-up studies, have consistently found evidence linking arsenic in drinking water to diabetes [109–112]. In addition, more recent studies have utilized larger sample sizes, refined measures of exposure and outcome, and advanced statistical techniques, while also adjusting for potential confounding factors including, but not limited to, age, sex and lifestyle. These and other examples of the epidemiological evidence supporting arsenic promotion of diabetes are summarized in Table 1. Despite the wealth of epidemiological evidence, additional research is still needed to elucidate the underlying biological mechanisms by which arsenic exposure might contribute to the onset and progression of diabetes, which is discussed in more detail below.

Table 1. Epidemiological evidence supporting arsenic promotion of diabetes.

| Country | Study Population | Age | Adjustments | Duration | As Concentration (In ppb or ppm) | Diabetic Assessment/Methods of Detection | Ref. |
|---|---|---|---|---|---|---|---|
| *Bangladesh* | 140 diabetic vs. 180 non-diabetic controls recruited with HbA1c level > 7% | ≥20 years | Age, sex, family history of diabetes, smoking habit, betel nut chewing, education | 2010 | 69.3–100.9 ppm in drinking water for 9.8–13.6 years | FBG ≥ 200mg/dL | [113] |
| | 115 exposed subjects diagnosed as arsenicosis patients (>50 μg/L As water consumption and skin lesions) and 120 unexposed volunteers | 14–85 years | Age, height and body weight | 2001–2003 | drinking water (0.218 ppm) and spot urine (20.235 ppm) | FBG ≥ 140 mg/dL | [114] |
| | 163 subjects with keratosis exposed to arsenic and 854 unexposed individuals | >30 years | Age, sex and body mass index | NR | 0.01–2.1 ppm in drinking water | history of symptoms: previously diagnosed diabetes, glycosuria and blood sugar level after glucose intake (OGTT) | [115] |
| | 1595 subjects depending on drinking water from wells: 1841 drank arsenic-contaminated drinking water but 114 had not | ≥30 years | Age, sex and body mass index | NR | well water > 0.05 ppm | Glycosuria | [116] |
| | 40 workers occupationally exposed to arsenic, 26 without any known As exposure and 6 who directly handle As containing products | 20–60 years | Sex, occupation, age, smoking habit | NR | 22.3–294.5 nmol per mmol of creatinine in urine sample of the exposed group | glycosylated hemoglobin (HbA1c) 5.4% compared to reference group 4.4% | [117] |
| *Chile* | population based cancer case-control study of 1301 participants in Northern Chile | ≥25 years | Age, sex, race, hypertension, cancer, socioeconomic status, smoking status | 2007–2010 | >0.8 ppm arsenic water concentration | physician diagnosed diabetes or oral hypoglycemic medication use | [118] |

**Table 1.** *Cont.*

| Country | Study Population | Age | Adjustments | Duration | As Concentration (In ppb or ppm) | Diabetic Assessment/Methods of Detection | Ref |
|---------|------------------|-----|-------------|----------|----------------------------------|------------------------------------------|-----|
| *China* | 2090 women with singleton pregnancy from the Tongji Maternal and Child Health Cohort (TMCHC) | ≥25 years | Pregnancy, education, income, ethnicity, fetal sex | 2013 | 0.3 ppb | Urine samples and oral glucose tolerance test, FBG ≥ 92 mg/dL | [119] |
|  | 335 gestational diabetes mellitus and 343 controls without GDM based on a prospective cohort established in Beijing, China | <35–≥35 years | Age, ethnicity, education, occupation, | 2017–2018 | 220 ppm | FBG ≥ 5.1 mmol/L, maternal hair samples | [120] |
|  | 1527 pregnant women drawn from Mother and Child Microbiome Cohort (MCMC) study | <30–≥30 years | Education, BMI | 2017–2018 | 0.83 ppb | 75-g oral glucose tolerance test (OGTT), FBG ≥ 5.1 mmol/L, 1 h postprandial ≥ 10.0 mmol/L, or 2 h postprandial glucose ≥ 8.5 mmol/L | [121] |
|  | 3474 women who were part of the Ma'anshan Birth Cohort (MABC) Study conducted from the City of Ma'anshan, Anhui Province of China | ≤24 years, 25–29 years, ≥30 years | Maternal age, BMI, gravidity, parity, income, education | 2013–2014 | 0.0047 ppb | FBG ≥ 5.1 mmol/L;1 h, ≥10.0 mmol/L; or 2 h, ≥8.5 mmol/L | [122] |
| *Croatia* | 202 adult urban participants from the city of Osijek in eastern Croatia and city of Zagreb in western Croatia | ≥45 years | Age, gender, education, smoking, family history if diabetes, physical activity, dietary consumption, origin of water used for drinking | 2018 | 0.5–361 ppb total urine As | FBG ≥ 3.5 mmol/L, HbA1c ≥ 37 mmol/L, insulin ≥ 15 pmol/L | [123] |

Table 1. *Cont*.

| Country | Study Population | Age | Adjustments | Duration | As Concentration (In ppb or ppm) | Diabetic Assessment/Methods of Detection | Ref |
|---|---|---|---|---|---|---|---|
| *India* | Natives to Nallampatti, an agricultural village in south India and part of the KMCH-NNCD cross-sectional study | ≥20 and ≤85 years | Age, sex, alcohol intake, smoking, tobacco use, BMI, education, occupation, familial diabetic history | 2015 | 4.10–63.30 ppm creatinine units of arsenic | blood investigation included a random glucose, HbA1c, cystatin-c, non-fasting lipid profile, uric acid and hemoglobin | [124] |
| *Italy* | 3390 art glass workers employed in 17 industrial facilities for at least 1 year | <40, 40–65 and >65 years | Age, sex, history of disease/mortality | 1950–1985 | 3.26 ppb in glassworks (>10 $\mu g/m^3$ in glassworks) | All causes of death coded according to the 8th revision of the ICD | [125] |
| | 258 subjectswith a minimum of two-year residency in the regions and without occupational exposure to As | ≥5 years | Age, sex, source of drinking water | 1993–2008 | 3–215 ppb iAs in drinking water, 2.3–233.7 ng/mL tAs in Urine | FBG ≥ 126 mg/dL, OGTT ≥ 200 mg/dL, HbA1c levels > 7%, self-reported diagnosis, or medication | [103] |
| | 200 diabetic cases and 200 controls | ≥30 years | Age, height, weight, body mass index, smoking habit, family history of diabetes, employment, location | 1960 | intermediate total As concentration in urine (63.5–104 $\mu g/g$ creatinine) | FBG ≥ 126 mg/100 mL (> or =7.0 mmol/l) or a history of diabetes treated with insulin or oral hypoglycemic agents | [126] |
| | 1160 adults with a minimum 5 year residency in study area | ≥18 years | Age, gender, ethnicity, education/occupation, smoking status, alcohol consumption, recent seafood intake, drinking water sources (well, treatment plant or other) and use and medical history | 2008–2013 | <0.01–419.8 ppb As in drinking water, tAs 0.52–491.5 ppb in urinary As. | FBG ≥ 126 mg/dL, 2HPG ≥ 200 mg/dL, self-reported diagnosis, or medication | [127] |

**Table 1.** *Cont*.

| Country | Study Population | Age | Adjustments | Duration | As Concentration (In ppb or ppm) | Diabetic Assessment/Methods of Detection | Ref |
|---------|-----------------|-----|-------------|----------|----------------------------------|------------------------------------------|-----|
| | 49 healthy individuals and 77 patients | NR | Age, sex, geographical location history of disease | NR | 0.32–9.82 ppb As in diabetic patients, mean As 3.44 ppb | Urine samples of diabetic patients to test As concentration | [128] |
| | 1451 randomly selected participants from Spain (representative sample of a general population) | ≥20 years | Age, sex, somking status, education, seafood consumption | 2001–2003 | 3.8 ppb of total plasma As, 106,000 ppb of total urine As, 14,900 ppb µg/g of iAs and 66,500 ppb of Asb in participants with diabetes | FBG ≥ 126 mg/dL and glycosylated hemoglobin (HbA1c) level > 6.5% or physician diagnosis or glucose lowering medication use | [129] |
| *Sweden* | 43 smelter workers exposed to iAs dust for 13–45 years | 44–70 years | age, height, smoking habit, alcohol consumption | 1987 | 1.6–63 ppb As in work-room air at the smelter | self-reported type 2 diabetes | [130] |
| | 12 cases with DM on death certificate and 31 controls employed in a Swedish copper smelter | 30–74 years | Age, history of diseas/death | 1960–1976 | <0.5–>0.5 ppb As | death certificate, medical record | [131] |
| | 5498 art glass workers in southeastern Sweden | ≥45 years | Age, occupation (glassworkers vs. glassblowers, other foundry workers and unspecified glass workers) | 1950–1982 | <1.9 ppb As in Swedish glassworks; <6 µg/m³ As in Swedish glassworks | All causes of death coded according to the 8th revision of the ICD | [132] |
| *Taiwan* | 891 adults in southern Taiwan village where arseniasis if hyperendemic | ≥30 years | Age, sex, body mass index, activity level at work | 1960–1970 | 0.1–15 ppm-year or higher | oral glucose tolerance test (OGTT) or self-reported history of diabetes treated with sulfonylurea or insulin | [104] |
| | Cancer and noncancer diseases | All age group | Sex, Age | 1971–1994 | 0.25–1.14 ppm As in artesian well water | All causes of death coded according to the 8th or 9th revision of the ICD | [105] |

**Table 1.** *Cont.*

| Country | Study Population | Age | Adjustments | Duration | As Concentration (In ppb or ppm) | Diabetic Assessment/Methods of Detection | Ref |
|---|---|---|---|---|---|---|---|
| | 446 nondiabetic residents in a village in Taiwan | ≥30 years | Age, body mass index and cumulative arsenic exposure | 1988–1989 | median As of artesian well water from 0.7 to 0.93 ppm | FBG ≥ 7.8 mmol/L and/or a 2 h post-load glucose level > or = 11.1 mmol/L. | [133] |
| | 66,667 residents living in endemic areas and 639,667 in nonendemic areas | ≥25 years | Age, sex | 1999–2000 | artesian well water > 0.35 ppm | All causes of death coded according to the 9th revision of the ICD (ICD-9 code 250 and A181) | [134] |
| | 4 townships in southwestern Taiwan where blackfoot disease is endemic | NR | Age, Sex | 1971–2000 | arsenic concentration of artesian well water ranged from 0.35 to 1.14 ppm with a median of 0.78 ppm | All causes of death coded according to the 8th or 9th revision of the ICD (ICD-9 code 250). | [135] |
| | 1297 subjects from an arsenicosis endemic area in southwestern Taiwan | ≥40 years | Age, sex, smoking status, education, exercise, alcohol consumption, betel nut intake | 1990, 2002–2003 | 0.7–0.93 ppm As in well water | FBG, cholesterol, triglycerides, low and high density lipoproteins, urine acid and urine creatinine levels, arsenic methylation patterns and GSTO1 genotypes linked to metabolic syndrome as an early factor for diabetes | [136] |
| UK | 32 insulin treated (ITDM), 55 non-insulin treated (NITDM) diabetic patients and 30 nondiabetic individuals (C-DNM) from Oxford, England | 18–78 years | Age, body mass index, glucose, insulin | NR | 0.018–0.2 ppm As | Glucose levels and insulin treatment | [137] |
| USA | 4549 American Indian participants | 45–75 years | Age, sociodemographic, smoking and alcohol status, height, weight, blood pressure | 1989–1991, 1998–1999 | 5.9–14 ppm iAs 14.3 ppb in Arizona, 11.9 ppb in Dakota, 7 ppb in Oklahoma | FBG ≥ 126 mg = dL, 2HPG ≥ 200 mg = dL, self-reported diagnosis, or medication | [138] |

Table 1. *Cont*.

| Country | Study Population | Age | Adjustments | Duration | As Concentration (In ppb or ppm) | Diabetic Assessment/Methods of Detection | Ref |
|---|---|---|---|---|---|---|---|
| | 1393 smelter workers | <20–40+ | Age, sex, race, occupation | 1946–1977 | 0.5–5 ppb As of air concentration in the insecticide building | All causes of death coded according to ICD | [139] |
| | 8014 copper smelter workers in Montana | <20–≥30 | Sex, Race | <1957, 1938–1989 | 0.29–11.3 ppb of airborne As | All causes of death coded according to the 8th or 9th revision of the ICD (ICD-8 codes 460–519) | [140] |
| | 1827 boys and 1305 girls | 2–14 years | Age, sex | 1907–1932 | 140–1600 ppm soil As concentration | All causes of death coded according to death records from the National Death Index, ≥47 and from Washington State (1900–1990), Oregon State (1971–1979), and California State (1960–1990), to locate deaths of cohort members | [141] |
| | Historical ward membership records of the Church of Jesus Christ of Latter-day Saints (LDS) (also known as the Mormons) | <50–80+ | Age, sex | 1977 | mean As 150 ppb, median As 14 to 166 ppb | Death certificate, mortality from hypertensive heart disease | [142] |
| | 1185 respondents from 19 townships in arsenic contaminated area | ≥35 years | Age | 1992–1993 | 2–>10 ppb As, with a median of 2 ppb As | Self reported | [143] |
| | 788 adults aged 20 years or older who participated in the 2003–2004 National Health and Nutrition Examination Survey (NHANES) and had urine arsenic determinations | ≥20 years | Age, sex, race, ethnicity; educational, smoking and alcohol consumption status; and dietary recall | 2003–2004 | 7.1 ppb total As, 3 ppb dmAs, 0.9 ppb arsenobetaine | FBG ≥ 126 mg/dL, self-reported physical diagnosis or use of insulin/oral hypoglycemic medication | [144] |

**Table 1.** *Cont.*

| Country | Study Population | Age | Adjustments | Duration | As Concentration (In ppb or ppm) | Diabetic Assessment/Methods of Detection | Ref |
|---|---|---|---|---|---|---|---|
| | 3925 people on tribal tolls in 13 American Indian communities | <55–≥65 | Age, sex, education, body mass index, smoking status, alcohol consumption | 1989–1991 | 7.9–24.2 ppb urine As, median urine As 14.1 ppb | Glycated hemoglobin and insulin resistance, fasting glucose level of 126 mg/dL or higher, 2 h glucose levels of 200 mg/dL or higher, hemoglobin A1c (HbA1c) of 6.5% or higher, or diabetes treatment | [145] |
| | cohort of American Indians in Arizona, Oklahoma, North Dakota and South Dakota | ≥30 years | Age, ancestry, family relationships | 1998–1999, 2001–2003, 2005–2006, 2014–2015 | median exposure of 5.93 ppb | FBG ≥ 126 mg/dL, or use of insulin or oral hypoglycemic medications | [109] |
| | non-institutionalized civilian resident population from NHANES | ≥20 years | Body mass index, age, gender, race/ethnicity, education, income, cigarette use, alcohol intake and physical activity | 2011–2014 | 246–260.6 ng/h | Spot urine samples, FBG ≥ 100 mg/dL or use of medication to treat hyperglycemia | [146] |
| | 4549 members of 13 tribes based in Arizona, Oklahoma, North Dakota and South Dakota | 45–75 years | Age, sex, study region, medical history, smoking status | 1989–ongoing | 10.2–11.2 nmol per mmol of creatinine in urine sample of the exposed group | Urinary arsenic species measured using HPLC to identify differentially methylated position | [110] |
| | 2919 participants recruited by Strong Heart Family Study | ≥25 years | Age, sex, education, smoking history, alcohol use, medical history | 1998–1999, 2001–2003 | median 0.52 ppb | Urine arsenic, FBG ≥ 126 mg/dL, self-reported physician diagnosis or self-reported use of insulin or oral diabetes treatment | [111] |

Table 1. *Cont.*

| Country | Study Population | Age | Adjustments | Duration | As Concentration (In ppb or ppm) | Diabetic Assessment/Methods of Detection | Ref |
|---|---|---|---|---|---|---|---|
| | Pregnant women with and without GDM who received prenatal care at the University of Oklahoma Health Sciences Center (OUHSC) Women's Clinic and High Risk Pregnancy Clinic | ≥18 years | Maternal age, race/ethnicity, education, income, history of GDM diagnosis | 2009–2010 | 1.25 ppb total arsenic | BG ≥ 135 mg/dL | [147] |
| | 688 participants including type 1, type 2 and control participants from SEARCH, a study being conducted in South Carolina, Colorado and Columbia | 10–22 years | Age, sex, race, education, height, weight | 2003–2006 | 0.0429–0.0502 ppb iAs | Clinical diabetes assigned by the health provider | [148] |
| | 5114 African-American and white men and women who are part of the CRADIA study living in Birmingham, AL; Chicago, IL; Minneapolis, MN; and Oakland, CA | ≥25 years | Age, gender, race, education, smoking status, alcohol consumption, physical activity, BMI, dietary intake | 1987–88; 2015–2016 | <0.0593–≥0.1692 ppm toenail arsenic level | fasting glucose ≥ 126 mg/dL, non-fasting glucose ≥ 200 mg/dL, 2 h postchallenge glucose ≥ 200 mg/dL, hemoglobin A1c ≥ 6.5%, or use of glucose-lowering medications. | [112] |

### 4.3. Mechanisms Associated with iAs-Induced Diabetogenesis

Epidemiological studies have revealed a greater incidence of diabetes among residents in areas highly contaminated with iAs, including Bangladesh [107], Taiwan [133], and Mexico [103]. These epidemiological studies in iAs-exposed populations clearly demonstrate an association between iAs and the pathological progression of DM. Along with the epidemiological evidence, laboratory studies have also shown that exposure to iAs can produce effects that correspond to diabetic phenotypes.

Despite a vast wealth of epidemiological correlations, as well as in vivo and in vitro experimental determinations of iAs-promoted diabetic phenotypes, mechanistic insight has remained limited. A variety of mechanisms for arsenic's diabetogenic effects have been proposed and demonstrated across a variety of tissue types and diabetic contexts. However, the exact mechanism for iAs-induced diabetic effects is still a matter of debate. Studies conducted thus far have implicated inhibition of insulin-dependent glucose uptake, pancreatic β-cell damage and/or dysfunction, and stimulation of hepatic gluconeogenesis as some of the major mechanisms involved in iAs-induced diabetes [125]. At the transcriptional level, other potential mechanisms of iAs-induced dysfunction include modulation of expression of genes involved in insulin signaling [149,150], as well as influencing adipocyte differentiation [151,152] (Figure 2). Thus, arsenic exerts its pro-diabetogenic effects by affecting multiple organ systems, diminishing their function over time.

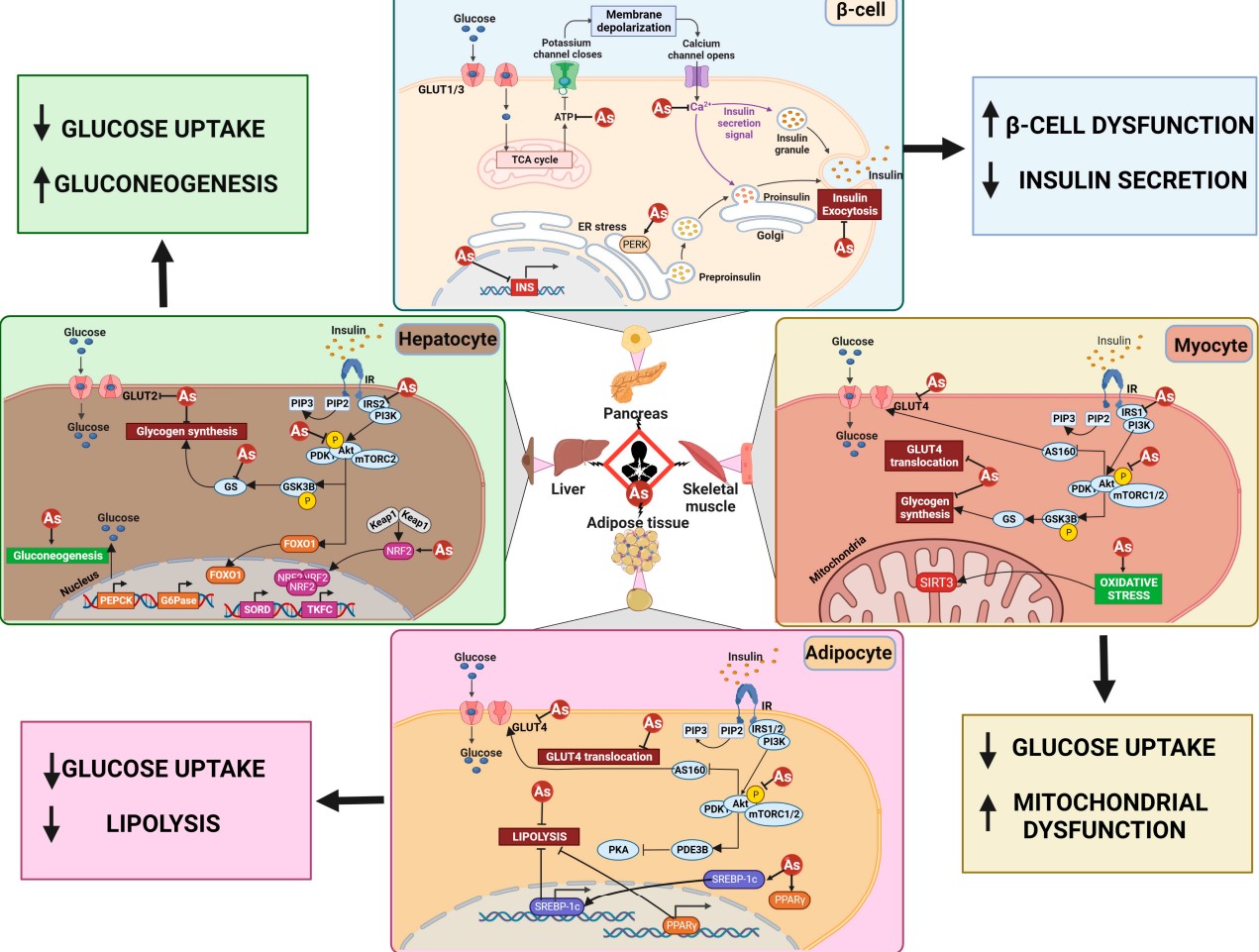

**Figure 2.** A variety of mechanisms have been proposed for the diabetogenic effects of arsenic. Shown here are (**top left**) stimulation of hepatic gluconeogenesis; (**top right**) a decrease in insulin secretion from beta cells; (**bottom left**) decreased glucose uptake and lipolysis in adipocytes; and (**bottom right**) decreased glucose uptake due to increased mitochondrial dysfunction.

Specifically, in vitro and in vivo studies have shown iAs-dependent inhibition of glucose transporter 4 (GLUT4) recruitment to the plasma membrane either directly or through inhibition of Akt, a key signaling enzyme required for GLUT4 translocation [153,154]. Arsenic can also play a role in decreasing the phosphorylation of mechanistic target of mTOR and p70, key regulators of insulin-stimulated glucose uptake [155]. In addition to inhibiting insulin signaling, iAs also stimulates hepatic gluconeogenesis by inducing the increased expression of phosphoenolpyruvate carboxykinase (PEPCK), a rate-limiting enzyme in gluconeogenesis, thus resulting in hyperglycemia even under fasted conditions [156,157]. Studies have also linked chronic iAs exposure to impaired pancreatic β-cell function, as higher blood glucose levels result in an increased demand on β-cells to produce more insulin, leading to their dysfunction over time [158]. Arsenic exerts its diabetogenic effects on skeletal muscle function through induction of oxidative stress and disruption of calcium homeostasis [159,160]. Arsenic induces oxidative stress in skeletal muscle by inhibiting enzymes involved in oxidative phosphorylation, resulting in decreased ATP production and increased oxidative stress [161]. This increases production of reactive species inhibits GLUT4 translocation, and interferes with the Akt pathway, leading to decreased glucose uptake in skeletal muscle [162]. In adipose tissue, chronic iAs exposure is also known to contribute to the development of obesity and other metabolic disorders through induction of oxidative stress, as well as disruption of adipokine signaling, and dysregulation of lipid metabolism [163]. Arsenic exposure can also decrease PDE3b (phosphodiesterase 3b) expression and activity, an enzyme that regulates lipolysis and glucose uptake in adipocytes, resulting in hyperglycemia and insulin resistance [164]. In addition, SREBP (a transcription factor that regulates lipid metabolism) and PPARg (a nuclear receptor that regulates adipogenesis and glucose metabolism) have both been shown to be activated by iAs in adipocytes, resulting in increased expression of lipogenic genes and adipogenesis, eventually leading to the development of obesity, insulin resistance, and hyperglycemia [164–166].

In the liver, iAs exposure can have harmful effects on hepatocytes by altering hepatic gene expression and signaling pathways involved in liver metabolism. For example, prolonged, non-canonical activation of the transcription factor NRF2 (nuclear factor erythroid 2-related factor 2), which results from autophagy inhibition, and p62-dependent sequestration of Keap1, the negative regulator of NRF2, has been shown to mediate insulin resistance and glucose intolerance in wild-type mice exposed to 25 ppm iAs for 20 weeks [167]. Besides NRF2, iAs has also been shown to influence the expression of other transcription factors that may be related to enhanced diabetes risk [168,169]. Chronic iAs exposure increased the gene expression of *PEPCK* (phosphoenolpyruvate carboxykinase) and *G6PC1* (glucose-6-phosphatase), two key gluconeogenic enzymes that promote hepatic glucose synthesis and thus contribute to hyperglycemia, via prolonged activation of the transcription factor FOXO1 (forkhead box O1) [170]. Exposure to iAs also increased *SORD* (sorbitol dehydrogenase), *TKFC* (transketolase-like protein 1), and *KHK* (ketohexokinase) expression in the liver, leading to increased hepatic glucose production via the polyol pathway, ultimately contributing to hyperglycemia in mice [167]. Overall, iAs exposure has been shown to induce diabetogenesis through multiple tissue-specific mechanisms.

In terms of acute arsenic iAs toxicity, including its effects on glucose metabolism, the binding of iAs to thiol (SH) groups has been shown. The reactivity of iAs on sulfhydryl groups can inactivate over 200 enzymes, and thus could be responsible, at least in part, for the widespread pathogenic effects of iAs on different organ systems [171,172]. Arsenic, in its trivalent form ($As^{3+}$), is also known to inhibit pyruvate and α-ketoglutarate dehydrogenase during acute poisoning, both of which are essential enzymes for gluconeogenesis and glycolysis [171]. In its pentavalent form ($As^{5+}$), it can substitute for phosphate, disrupting protein phosphorylation and oxidative phosphorylation [173]. However, whether this occurs in a chronic exposure context, as well as at more physiologically relevant concentrations, has yet to be determined.

Increasing iAs levels in the blood correlated with increasing levels of ROS and decreased antioxidant capacity in the plasma of iAs-exposed individuals in Taiwan, suggesting

the influence of the iAs-ROS axis on promoting diabetes [174]. Oxidative stress, inflammation, and apoptosis have all been implicated as pathways that could converge to link iAs exposure with DM onset and progression [175]. These mechanisms fit arsenic's effects on systemic metabolism, as in normal mice, iAs exposure has been shown to result in prediabetic effects via alterations to lipid metabolism, gluconeogenesis, and insulin secretion, while also worsening diabetic outcomes in a diabetic mouse model [156]. Thus, the ability of arsenic to dysregulate these processes involved in both early and later outcomes associated with DM, establishes iAs as a relevant diabetogen.

Finally, iAs is also known to impact various components of the epigenetic machinery. Exposure to iAs has been linked to varied gene expression of *AS3MT* [153], *CAPN10* [158], *GSTO1* [136] and *NOTCH2* [176]. Differences in genotype, as well as single-nucleotide polymorphisms (SNPs) in any of these critical genes, can dictate the risk of developing DM, as the iAs metabolite profile, as well as glucose metabolism, can vary greatly. Supporting this notion, studies in both human cohorts and in vivo experimental models have shown iAs-induced changes in epigenetic regulation of glucose homeostasis, specifically DNA methylation and miRNA suppression of DM-related genes involved in glycemic regulation [177,178]. Furthermore, iAs-associated changes in DNA methylation of DM-related genes were observed in the peripheral blood leukocytes of individuals consuming high levels of iAs in the drinking water in Mexico [179]. At the miRNA level, a study examining newborn umbilical cord blood samples for miRNA expression following in utero iAs exposure indicated altered expression of miR-107 and miR-20b, both of which have been associated with DM [180]. Similar in vivo studies in the liver tissue of mice exposed to various concentrations of sodium arsenite also revealed altered miRNA expression profiles [181].

*4.4. Future Research Needs*

Based on the evidence described above, it is clear that arsenic toxicity is dependent on exposure dose, frequency, duration, and species involved, as well as the age, gender, and individual genetic susceptibilities of the exposed individual, amongst many other variables [182]. Several of these parameters should be further explored in future studies to determine the association between iAs toxicity in drinking water and the progression of DM. Specifically, the epigenetic aspect of iAs-controlled diabetes induction remains understudied, which could provide key insight into understanding this aspect of iAs promotion of diabetes, particularly when changes in diabetes-relevant gene expression are observed. Dietary influences and genetic polymorphisms in response to iAs exposure should also be further studied, as they could provide key insight into how different regional populations are affected during exposure. Investigating the role of iAs in dictating diabetogenic changes at the cellular level also requires more experimental evidence using consistent and exposure-relevant doses of iAs. Improved consistency and dose relevance, coupled with the identification of appropriate biomarkers for iAs-induced DM, will allow for a better comparison amongst exposed and unexposed groups.

Another important issue moving forward is that the conditions of exposure to iAs in humans overall need to be more fully characterized so that better biomarkers can be developed and the separation of relevant forms of iAs and their level of toxicity at the tissue vs. systemic level can be better defined. The bioaccumulation of various forms of iAs in cellular versus animal models and their relevance to human physiological settings is also therefore considered an important area for further research. Gender and age differences in susceptibility to iAs and their relation to development of diabetes are also poorly defined. In addition, metal–metal interactions should also be studied to define the consequences of iAs interaction with other harmful metals, as arsenic is not the only toxic component present during exposure. Altogether, while much is known epidemiologically regarding the increased risk of diabetes associated with chronic arsenic exposure, a great deal still needs to be done at the experimental level to increase our understanding of arsenic's diabetogenic effects and generate relevant therapies for this subset of diabetic patients.

## 5. Regulation of Arsenic in Drinking Water

Disrupting exposure to arsenic requires understanding of water sources that may contain elevated levels, whether naturally occurring or as a result of contamination. Monitoring arsenic in drinking water is critical, especially for those who are on private wells. Private well-water quality outreach and sampling campaigns have been conducted across the country to protect human health and address arsenic exposure. For example, the collaborative public health project "All About Arsenic," was initiated in 2015 by researchers at Mount Desert Island Biological Laboratory and Dartmouth College's Toxic Metals Superfund Research Program to "to expand private well water testing for arsenic and other elements and to build data literacy among students and the wider public" [183,184]. A key factor to ensure the success of these monitoring and educational programs is public participation, research conducted with nonprofessionals, who may contribute to the research question, generation of theory or hypothesis, data collection, data analysis, data interpretation, and/or translating research to action (e.g., [185]). Public participation in research is a valuable model for investigations across disciplines and can connect science and practice to people and policy (e.g., [186]). Practices led by institutions only have been critiqued for their lack of accessibility, diversity, justice, equity, and inclusion [187]. Community-based participatory research and community science efforts that champion placed-based topics and local experts and address community questions are strongly recommended and can increase the rigor and relevance of the effort (e.g., [187–189]). For example, Gardenroots [31,190–193], established in 2010, revealed that in one community, the local water utility was serving water that exceeded the arsenic drinking water standard (0.010 mg L$^{-1}$) [190]. Gardenroots participants worked together to identify and notify additional households that were connected to the public water supply. They also reported their test results to USEPA and Arizona Department of Environmental Quality, advocating that this issue needed to be addressed (Gardenroots also notified and sent the results to the USEPA). As a result, the municipal water suppler was issued seven notices of violation by the ADEQ, one for exceeding the arsenic drinking water standard. Additionally, arsenic concentrations in private well water exceeded the drinking water standard for several participants who relied solely on this water source. University of Arizona researchers worked closely with those households to provide information regarding water treatment technologies that could be implemented to reduce their arsenic concentrations [190].

## 6. Approaches to Removal of Arsenic from Drinking Water

The present USEPA national primary drinking water regulations (NPDWS) limit for arsenic is 10 µg/L. This is a compromise between consumer health protection and water treatment costs, as the USEPA has set a public health goal of arsenic in drinking water at "zero." Arsenic is found in drinking water in two forms of inorganic arsenic, although organic forms of arsenic also exist and can be found in aquatic environments such as benthic sediments. The common forms of inorganic arsenic include arsenate ($As^{+5}$) and arsenite ($As^{+3}$) (Figure 1). The more oxidized arsenate ions predominate in moderately to well-aerated water sources, whereas arsenite forms predominate in organic matter-rich, oxygen-limited waters. A 2014 study of 65 drinking water wells from 28 states in the US [194] showed that either arsenate or arsenite predominated in 91% of the wells, while the remaining wells had a combination of the two arsenic forms. The 91% of wells with a dominant arsenic form were distributed approximately evenly between arsenic and arsenate.

Although arsenate and arsenite are known to have different toxicities, the USEPA only monitors and regulates arsenic cumulatively in its elemental form (As). The amount of arsenic in surface and groundwater depends primarily on the surrounding geology, as well as industrial activity, including, among others, mining and oil extraction. Arsenic is commonly associated with pyritic (iron- and sulfur-containing) minerals, which when exposed to oxidizing–acidic conditions release arsenic in the water environment as arsenate

and/or arsenite ions. A common arsenic mineral is arsenopyrite, often found with other pyritic minerals rich in copper, lead, cadmium and other metals.

Lowering the levels of arsenic in drinking water is difficult due to the complex chemistry of this element. Although the arsenic in arsenic-rich minerals is relatively insoluble in natural waters (except in extreme redox and pH conditions), areas with high amounts of arsenic-containing minerals often have naturally high levels of dissolved arsenic in groundwater. Lowering the levels of this element to drinking water standards can be difficult and expensive due to its shifting chemical forms. For example, changing the water redox potential or pH conditions can lead to the precipitation of arsenate and arsenite with iron, calcium and other cations leading to the formation of secondary arsenic-rich minerals. The pH range of most potable water sources is 6 to 9, which when combined varying oxygen levels can lead to the presence of arsenate or arsenite as previously described. These arsenic species have difference sizes, charge (-), and reactivity, complicating their removal from water using precipitation, absorption, ion exchange, and nanofiltration processes used in today's best available water treatment technologies. The next sections present a summary of treatment technologies that can be used to lower arsenic levels for both public utilities and home water treatment systems.

The USEPA has guidelines and recommendations for the selection of best-available technology (BAT) to mitigate arsenic in water, given variables such as the number of connections (consumers), water quality, levels of arsenic in water, location, infrastructure, etc. [195]. Public water utilities should follow these guidelines in the selection and testing of the BAT or BATs to ensure consistent compliance to the arsenic standard at the lowest cost to the consumer. There is sometimes tension between the choice of arsenic levels and cost of treatment [194].

### 6.1. Technologies for Public Water Utilities

#### 6.1.1. Blending

Mixing two water sources to produce water with arsenic levels below the NPDWS is an acceptable technology available to water utilities with diverse sources of potable water such as surface water and groundwater [196].

#### 6.1.2. Coagulation/Filtration

The addition of iron salts such as ferric chloride or sulfate to well-aerated water leads to the formation of insoluble amorphous ferric hydroxides that adsorb preferably arsenate anions entrapping them into a coagulant that can settle and be filtered out of the water. The efficiency of the treatment process can be optimized up to 95% by adjusting the pH with the proper selection of iron salts and the addition of oxidizing agents such as chlorine and permanganate to oxidize arsenite to arsenate. This treatment technology produces significant amounts of potentially hazardous arsenic-contaminated residues that must be disposed of (usually landfilled) following federal and state guidelines [195].

#### 6.1.3. Oxidation/Filtration

Oxygen-free groundwater may have significant amounts of soluble iron and/or manganese present, often accompanied by soluble arsenite. In this case, water aeration or the addition of an oxidizing chemical leads directly to the formation of both arsenate and insoluble ferric hydroxides that can sorb the arsenate. This is followed by filtration to remove iron–manganese–arsenic particles. The efficacy of this approach depends on the initial ratio of iron to arsenic present in the water. This technology also requires the proper disposal of arsenic-contaminated residues [195].

#### 6.1.4. Metal Oxides

Since arsenic anions have a high affinity for positively charged metal oxides, adsorptive materials composed of solid porous media such as aluminum oxides (activated alumina) and many types of ferric hydroxy-oxides (GFH) (alone or coated onto inert solid media)

are options for closed water treatment systems. These approaches can efficiently filter out arsenate with up to 95% removal, provided that all forms of arsenic are present as arsenate. This again may require the conversion of arsenite, if present, to arsenate with the addition of oxidants as well as pH adjustment to optimize arsenic removal efficiencies. Since there are many manufacturers of these materials and varying costs, pilot studies are usually required to test materials and determine best pre- and posttreatment(s) needed to optimize arsenic removal and lower costs. Importantly, knowledge of the water chemistry (e.g., salinity, pH, alkalinity, redox potential, and the concentrations of other potentially competing ions) is also needed [195]. The presence of other ions such as fluoride, silica, and sulfate can also interfere with the adsorption of arsenate. Once spent, these porous media must be disposed of as potentially hazardous arsenic contaminated residues.

Innovative particle coatings and nanoparticles made of and with carbon, alumina, iron, titanium, zirconium and other elements are being explored for As removal from water [197] with varying degrees of success, higher costs and remaining challenges associated with the disposal of spent materials.

### 6.1.5. Anion Exchange Resins

Porous synthetic organic polymer beats populated with positively charged sites saturated with a common anion such as chloride ($Cl^-$) can be used to efficiently remove arsenate ions from water. These resins can be manufactured to preferentially remove arsenate anions over other common anions as mentioned previously. Ion exchange resins are more expensive than inorganic porous media but have the advantage that they can be reused after regeneration with alkali solutions. Resin regenerant wastes containing arsenic must be disposed of as hazardous waste. As with other treatment technologies, pre- and posttreatment(s) may be necessary to oxidize any reduced arsenic forms to arsenate [195].

### 6.1.6. Enhanced Lime Softening

The addition of lime [$Ca(OH)_2$] to water is commonly used to reduce hardness through precipitation of calcium and magnesium. This technology can also be used to remove arsenic [198,199]. Lime is added to bring the pH of the system to higher than 10.5. This results in precipitation of carbonates ($CaCO_3$) and hydroxides [$Mg(OH)_2$], and when arsenate is present, it too will precipitate. Magnesium additions may be needed if not present in the water and posttreatment is required for pH adjustment. This process is quite efficient for arsenate, but less efficient for arsenite. The technology requires large amounts of lime, which in turn generates large amounts of waste sludge. In addition, the high operating pH can be problematic and the treated water needs pH adjustment following treatment [195].

### 6.1.7. Nanofiltration and Reverse Osmosis

These similar types of membrane filtration (adsorption of ions onto a semiporous membrane) processes are best suited and most cost-effective for treatment of water with total dissolved solids (TDS) greater than 500 mg/L and when other ions besides arsenic must be lowered to meet drinking water standards. A complete analysis of all major and minor water quality parameters must be performed for the initial evaluation and testing of these processes, since implementing either technology just to reduce arsenic levels in water would not be cost-effective. Note that reverse osmosis (RO) systems are more expensive to operate than nanofiltration, but are more efficient at lowering arsenate. Pressures from 50 to over 200 psi may be used to force influent water through a semiporous membrane, which produces scaling and fouling requiring periodic flushing. This membrane cleaning step can produce significant volumes of brackish water that must be disposed appropriately. Up to 70% of the influent water may be lost during the membrane cleaning cycle, depending on the levels of particulates, bacteria and salts present in the water [197]. For example, very hard water can increase membrane scaling significantly. As with previous technologies, the

arsenate forms are preferentially adsorbed (RO > 95%). Therefore, if arsenite is present in the influent, it must be converted to arsenate with pretreatment oxidation [195].

### 6.2. Home Treatments

Point of use (POU) devices are typically used by homeowners that have elevated levels of arsenic in their well water [200]. Today, homeowners have an increasing array of point of entry (POE) water systems to lower arsenic and other contaminants in water including water softeners, alkali, permanganate, chlorination, activated carbon, GFH, and RO systems, costing thousands of dollars to install and maintain. In the next sections we will summarize three low cost POU systems available to homeowners to lower arsenic levels in water.

### 6.2.1. Distillation

This process is straightforward, requiring the use of a steam-distilling unit that generates steam that when condensed produces contaminant-free, disinfected water. Tabletop steam-distillation units are slow and energy-intensive, producing enough water for daily drinking and cooking. During the distillation process, all arsenic forms and other ions present in water concentrate and precipitate, forming a scale in the distillation vessel that must be periodically cleaned out. Modern distillation units also have activated carbon filters that can trap volatile contaminants [200].

### 6.2.2. Reverse Osmosis

Small, under-the-sink, home water treatment systems that use the reverse osmosis process are widely available for do-it-yourself and professional installation. These small systems are fully automated and make use of the existing household water pressure (40–60 psi) to force water through a semipermeable membrane, storing it in a reservoir for later use. As with industrial systems, membrane fouling and scaling require periodic (and more frequent) washing. Up to nine volumes of water may be lost during this cycle for every volume of water produced depending on the influent water quality. High water TDS and hardness decrease RO system performance, significantly increasing household water consumption. In areas of the US with very hard water, a softening pretreatment may be needed. When used to lower arsenic or other primary contaminants, homeowners should test their water before and after RO treatment and regularly thereafter to make sure that arsenic or any other primary drinking water standards are met.

### 6.2.3. Iron Filters

Small POU in-line GFH filters are slowly becoming available to homeowners to filter arsenic out of water. However, their arsenic-filtering capacity is very much dependent on the concentrations of several other water ions commonly present in water, such as silica, sulfate, and fluoride, and other anions, as previously mentioned. Therefore, the homeowner should test arsenic levels in the treated water periodically to check the arsenic-removal efficiency of these filters over time. These filters do not generate any waste while in use, but they cannot be regenerated. Thus, when exhausted, they should be handled and disposed of as potentially hazardous materials [200].

### 6.3. Summary—Challenges to Removal of Arsenic from Drinking Water

The EPA lowered the arsenic drinking water standard of arsenic in 2001 from 50 μg/L to 10 μg/L. Ideally, drinking water should not have any arsenic, but reducing the levels of this element below 10 μg/L is difficult and expensive. This is because, as discussed above, arsenic exists in several forms in water and complex multistep treatments are often required to reduce arsenic levels in water. Because of its geological origins and arsenic's affinity for iron hydroxides and aluminum oxides (alumina), large-scale treatment has traditionally focused on the use of coagulation, coprecipitation, and sorption of arsenic using iron-based chemicals and alumina. Although new iron, other metal-based, and hybrid nanomaterials

(silica and activated charcoal with metal coatings and Fe, Ti, and Zr nanoparticles for example) are being developed for arsenic capture, their high cost and varying efficacies remain a challenge. Anion exchange resins remain an expensive but efficient method to lower arsenic concentrations in water. Membrane filtration systems such as nanofiltration are increasing in performance, with lower energy costs and lower levels of other water contaminants, such as salts, nitrate, metals, and many organic contaminants in addition to arsenic. For most of these approaches, an added concern is that the environmentally safe and cost-effective disposal of arsenic-contaminated solid and liquid residues, sorbents, and brines remains a challenge. Small-scale treatment systems such as POUs are increasing in popularity, in particular home RO systems. These systems also have associated costs due to maintenance requirements, increased water use, and water testing costs.

**Author Contributions:** Conceptualization, R.M.M., A.S. and M.D.; writing—original draft preparation, A.S., M.D., J.F.A., M.R.-A., R.A.R., J.C., R.M.M.; writing—review and editing, R.M.M., A.S., M.D., J.F.A., X.D.; visualization, A.S., M.D., R.A.R.; funding acquisition, R.M.M., X.D., J.C. All authors have read and agreed to the published version of the manuscript.

**Funding:** This work was supported by the National Institute of Environmental Health Sciences Superfund Research Program through grant P42ES004940.

**Institutional Review Board Statement:** Not applicable.

**Informed Consent Statement:** Not applicable.

**Data Availability Statement:** No new data were created or analyzed in this study. Data sharing is not applicable to this article.

**Conflicts of Interest:** The authors declare no conflict of interest.

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
