# Peer review of "Arsenic in Drinking Water and Diabetes"

_water, doi:10.3390/w15091751_

Round 1
Reviewer 1 Report
This review gives information about the distribution of arsenic in the environment, the relationship between arsenic and diabetes, and the removal of arsenic from drinking water. It was well organized, and comprehensive literature was cited. I have no specific comments. I recommend that the manuscript be accepted after addressing the following minor issues.
A substantial number of old literature were included in Table 1, and I suggested that recent research progress could be involved and compared with previous reports.
The "iAs" should be defined when it was used first.
In addition, a few input and format errors should be checked carefully.
Author Response
Reviewer 1:
This review gives information about the distribution of arsenic in the environment, the relationship between arsenic and diabetes, and the removal of arsenic from drinking water. It was well organized, and comprehensive literature was cited. I have no specific comments. I recommend that the manuscript be accepted after addressing the following minor issues.
A substantial number of old literature were included in Table 1, and I suggested that recent research progress could be involved and compared with previous reports.
We have updated the table to include more recent studies and written the following brief synopsis comparing previous reports to those that are more current:
“The number of epidemiological studies examining the relationship between diabetes and arsenic in drinking water has risen in recent years. These studies, which also include follow-up studies, have consistently found evidence linking arsenic in drinking water to diabetes (8,10,11,14). In addition, more recent studies have utilized larger sample sizes, refined measures of exposure and outcome, and advanced statistical techniques, while also adjusting for potential confounding factors including, but not limited to, age, sex and lifestyle. These and other examples of the epidemiological evidence evidence supporting arsenic promotion of diabetes are summarized in Table 1. Despite the wealth of epidemiological evidence, additional research is still needed to elucidate the underlying biological mechanisms by which arsenic exposure might contribute to the onset and proression of diabetes, which is discussed in more detail below.”
The "iAs" should be defined when it was used first.
This has been done, see Section 2.
In addition, a few input and format errors should be checked carefully.
The manuscript has been edited throughout for grammar etc.
Reviewer 2 Report
This review focuses on the most recent information regarding the distribution of arsenic in the environment and how it affects human health, particularly in relation to diabetes, which is one of the most significant chronic diseases, as well as the control of arsenic in drinking water and methods for treating it in public utilities. This review study also highlights the difficulties in comprehending the complex health effects of arsenic and in putting treatment plans into practice to significantly lower environmental exposures to arsenic. The authors have used very recent references to write this review. Although this review is extensive, the author should address the following minor comments.
- The authors should include references for technologies in the public water utilities section.
- Authors, please improve the magnification of figure 2.
- The authors should discuss more about signaling pathways related to diabetes.
Author Response
Reviewer 2:
This review focuses on the most recent information regarding the distribution of arsenic in the environment and how it affects human health, particularly in relation to diabetes, which is one of the most significant chronic diseases, as well as the control of arsenic in drinking water and methods for treating it in public utilities. This review study also highlights the difficulties in comprehending the complex health effects of arsenic and in putting treatment plans into practice to significantly lower environmental exposures to arsenic. The authors have used very recent references to write this review. Although this review is extensive, the author should address the following minor comments.
- The authors should include references for technologies in the public water utilities section.
The best, most up to date reference for this section is the USEPA 2022. We have cited this reference in the appropriate places now and added one further reference:
Nicomel N.R.; Leus K.; Folens K.; Van Der Voort P.; Laing G.D. (2016) Technologies for Arsenic Removal from Water : Curren Status and Future Perspectives. International J. Environ Res Public Health 2016 Jan; 13(1):62.
- Authors, please improve the magnification of figure 2.
An updated version Figure 2 with improved magnification has been included.
- The authors should discuss more about signaling pathways related to diabetes.
We have included a more detailed discussion of the diabetes relevant pathways affected by iAs exposure, including those relevant to Figure 2, in Page 12 under Section 4.3.
Reviewer 3 Report
The author's purpose of the review is interesting and timely, about “Arsenic in Drinking Water and Diabetes. The paper is globally well written, clear and easy to read and understand. I would recommend the suggestions described below:
1) At the introduction the authors referred that As is at the top of the ASTDR list. But from which year? Is it the second after Hg? Or the first in the last ten years? I am following that list since 1993. I think that the authors should clarify the position of As once it is the main purpose of the paper. References should be updated, for instance regarding heavy metals pollution with 2022 papers.
2) Also, it not clear at the end of the introduction what are the objectives of the paper, highlighting the contribution of the paper and what is timely and new.
3) At the introduction I would like to make the following suggestion: a figure or scheme with a short chronology or a short timelime of the major events in the field for instance, first studies, first in vivo studies, would be interesting and usual for a better understanding of the paper. Moreover, this short timeline will also reflect the understanding of the authors about the studies of As and diabetes. This personal view timeline will be interesting and also pedagogical for the others researchers being or not the field.
4) The figures are good but they could be globally improved, as possible, once Water deserves high quality figures and with rigor to avoid lacking of interest for the data.
5) At page 6 and section 4.1 the authors only refer to one reference. However, recent references about the topic could be inserted and discussed.
Author Response
Reviewer 3: The author's purpose of the review is interesting and timely, about “Arsenic in Drinking Water and Diabetes. The paper is globally well written, clear and easy to read and understand. I would recommend the suggestions described below:
- At the introduction the authors referred that As is at the top of the ASTDR list. But from which year? Is it the second after Hg? Or the first in the last ten years? I am following that list since 1993. I think that the authors should clarify the position of As once it is the main purpose of the paper. References should be updated, for instance regarding heavy metals pollution with 2022 papers.
Arsenic has been at the top of the ATSDR list since 1997. The paper has been edited to reflect this and a reference to the ATSDR list from 1997 has been added.
More recent citations (listed below) have been added to the section regarding heavy metals pollution:
“Metal(loid)s such as zinc (Zn), selenium (Se), copper (Cu), molybdenum (Mo), chromium (Cr), manganese (Mn), nickel (Ni), cobalt (Co), iron (Fe), magnesium (Mg), and arsenic (As) rank among the top priority metals that act as environmental toxicants in drinking water worldwide (WHO/FAO/OAEA, 1996; Tchounwou et al., 2012)”:
- Lotfi, S., Chakit, M., & Belghyti, D. (2020). Groundwater Quality and Pollution Index for Heavy Metals in Saïs Plain, Morocco. Journal of health & pollution, 10(26), 200603. https://doi.org/10.5696/2156-9614-10.26.200603
- Rehman, K., Fatima, F., Waheed, I., & Akash, M. S. H. (2018). Prevalence of exposure of heavy metals and their impact on health consequences. Journal of cellular biochemistry, 119(1), 157–184. https://doi.org/10.1002/jcb.26234
- Waseem, A., Arshad, J., Iqbal, F., Sajjad, A., Mehmood, Z., & Murtaza, G. (2014). Pollution status of Pakistan: a retrospective review on heavy metal contamination of water, soil, and vegetables. BioMed research international, 2014, 813206. https://doi.org/10.1155/2014/813206
- Bacquart, T., Frisbie, S., Mitchell, E., Grigg, L., Cole, C., Small, C., & Sarkar, B. (2015). Multiple inorganic toxic substances contaminating the groundwater of Myingyan Township, Myanmar: arsenic, manganese, fluoride, iron, and uranium. The Science of the total environment, 517, 232–245. https://doi.org/10.1016/j.scitotenv.2015.02.038
5. Tomlinson, M. S., Bommarito, P., George, A., Yelton, S., Cable, P., Coyte, R., Karr, J., Vengosh, A., Gray, K. M., & Fry, R. C. (2019). Assessment of inorganic contamination of private wells and demonstration of effective filter-based reduction: A pilot-study in Stokes County, North Carolina. Environmental research, 177, 108618. https://doi.org/10.1016/j.envres.2019.108618”
- Also, it not clear at the end of the introduction what are the objectives of the paper, highlighting the contribution of the paper and what is timely and new.
Paragraph is added at the end of introduction
- At the introduction I would like to make the following suggestion: a figure or scheme with a short chronology or a short timelie of the major events in the field for instance, first studies, first in vivo studies, would be interesting and usual for a better understanding of the paper. Moreover, this short timeline will also reflect the understanding of the authors about the studies of As and diabetes. This personal view timeline will be interesting and also pedagogical for the others researchers being or not the field.
We appreciate the reviewer’s interest in a timeline of arsenic-diabetes relevant discoveries; however, in this review we wanted to primarily provide an overview of the epidemiological and mechanistic studies supporting arsenic promotion of diabetic phenotypes, as well as the relevance of this topic to public health. We do not feel that the current state of the field/literature supports the generation of a definitive timeline of “key” or “relevant” discoveries, and as such decided to not include in this manuscript. Our hope is that continued development of the field will facilitate the generation of a similar figure in future reviews.
- The figures are good but they could be globally improved, as possible, once Water deserves high quality figures and with rigor to avoid lacking of interest for the data.
Higher quality figures submitted.
- At page 6 and section 4.1 the authors only refer to one reference. However, recent references about the topic could be inserted and discussed.
Additional references (indicated below) have been included as requested.
- American Diabetes Association; Diagnosis and Classification of Diabetes Mellitus. Diabetes Care 1 January 2004; 27 (suppl_1): s5–s10. https://doi.org/10.2337/diacare.27.2007.S5
- DeFronzo, R., Ferrannini, E., Groop, L. et al. Type 2 diabetes mellitus. Nat Rev Dis Primers 1, 15019 (2015). https://doi.org/10.1038/nrdp.2015.19
- Katsarou, A., Gudbjörnsdottir, S., Rawshani, A. et al. Type 1 diabetes mellitus. Nat Rev Dis Primers 3, 17016 (2017). https://doi.org/10.1038/nrdp.2017.16
- (21) Rehman, K., Fatima, F., Waheed, I., & Akash, M. S. H. (2018). Prevalence of exposure of heavy metals and their impact on health consequences. Journal of cellular biochemistry, 119(1), 157–184. https://doi.org/10.1002/jcb.26234”